# How Have Leukocyte In Vitro Chemotaxis Assays Shaped Our Ideas about Macrophage Migration?

**DOI:** 10.3390/biology9120439

**Published:** 2020-12-02

**Authors:** Agata N. Rumianek, David R. Greaves

**Affiliations:** Sir William Dunn School of Pathology, University of Oxford, Oxford OX1 3RE, UK; agata.rumianek@path.ox.ac.uk

**Keywords:** macrophage, chemotaxis, in vitro assays, transwell, leukocyte

## Abstract

**Simple Summary:**

The migration of immune cells is vital during inflammatory responses. Macrophages, which are a subset of immune cells, are unique in the ways they migrate because they can switch between different mechanism of migration. This crucial feature of macrophage migration has been underappreciated in the literature because technologies used to study macrophage migration were not able to efficiently detect those subtle differences between macrophages and other immune cells. This review article describes popular technologies used to study macrophage migration and critically assesses their advantages and disadvantages in macrophage migration studies.

**Abstract:**

Macrophage chemotaxis is crucial during both onset and resolution of inflammation and unique among all leukocytes. Macrophages are able to switch between amoeboid and mesenchymal migration to optimise their migration through 3D environments. This subtle migration phenotype has been underappreciated in the literature, with macrophages often being grouped and discussed together with other leukocytes, possibly due to the limitations of current chemotaxis assays. Transwell assays were originally designed in the 1960s but despite their long-known limitations, they are still one of the most popular methods of studying macrophage migration. This review aims to critically evaluate transwell assays, and other popular chemotaxis assays, comparing their advantages and limitations in macrophage migration studies.

## 1. Historical Overview

Cell migration (chemotaxis) is one of the most fundamental cell functions, especially during inflammation. The importance of pus, which we now know contains migrated leukocytes, was recognised by the ancient Romans [1]. Chemotaxis as a term was first used by Pfeffer in 1884 to describe the attraction of spermatozoids of ferns toward malic acid [2]. Although in 1841, Addison, and in 1873, Cohnheim, had observed cells emigrating from the blood into the inflamed tissue, it was Leber, in 1888, who noted the directionality of cell movement and proposed that leukocytes may also undergo chemotaxis. He injected guinea pig corneas with irritants and, after a few hours, observed leukocyte directional migration towards the injection site [3,4]. Building on that observation, Leber as well as others in the early 20th century were able to establish that leukocytes moved directionally towards bacteria and tissue breakdown products, a model widely accepted by the scientific community by the 1940s. In 1942, Wintrobe published the first edition of his *Clinical Haematology*, which contained the classical definition of neutrophil chemotaxis [1]. Most chemotaxis experiments were performed on translucent membranes in live animals (frog mesentery, tadpole tails, rabbit ear membranes) injected with starch or bacteria to induce chemotactic responses [1,3].

Because of these technical limitations, Harris challenged the whole field in the early 1950s by exposing the lack of appropriate controls and arguing that the previously observed accumulation of leukocytes at inflammation sites could be explained by the non-specific trapping of cells rather than directional migration. He pointed out the need for actual recordings of cell movement in an isolated system that would prove the directionality and specificity of the migration [3]. Harris designed a system where cells were trapped in a film of clotted plasma between a slide and a coverslip and colonies of bacteria were introduced. Using dark-field microscopy and long exposures of a single photographic film, he was able to track the granulocyte paths in the film through time and prove that cell movement is directional and induced by the addition of bacteria [5].

Harris’ chemotaxis assay was soon followed by a new “transwell” setup designed by Boyden in 1962, which simplified and standardised chemotactic experiments, and allowed studies of purified cell populations and quantitative analysis [6]. Boyden’s assay became the most popular chemotaxis assay, paving the way for the rapid expansion of cell migration studies. Modified versions of the original Boyden apparatus, called transwell assays, are the most widely used chemotaxis platforms despite minimal design changes since the 1960s.

## 2. Directional Cell Migration

Directional cell migration can be observed in all living organisms, from prokaryotic bacteria to mammals. Directional migration in mammals is fundamental during embryonic development, where it is involved in the large-scale migration of cells during gastrulation. In an adult organism, chemotaxis is involved in maintenance of homeostasis [7] but it can also be involved in pathological processes such as tumour metastasis, angiogenesis and atherosclerotic plaque development [8,9,10]. The migration of eukaryotic cells can be sub-divided based on how the cells respond to the signal, what shape changes the cells undergo or whether they migrate in single-cell or collective manner.

Single-cell migration is mostly prominent in the adult organism (e.g., keratinocytes, fibroblasts and leukocytes migration) [11]. During single cell-migration, every cell polarises and moves individually in response to a signal, and therefore requires both the receptors to respond to the signal and the expression of migratory machinery to perform intracellular reorganisation. On the other hand, collective migration occurs when cells interact with each other, both mechanically and chemically, to move as a coherent collective [12,13]. Collective migration is the predominant form of cell movement during development but has also been shown to contribute to cancer metastasis and wound repair [12]. To date, it has not been reported for any leukocyte population, and therefore will not be discussed further in this review.

### 2.1. Types of Directional Cell Migration: Signal-Dependent

Directional cell migration is often broadly called chemotaxis. However, this term is technically only applicable to the directional migratory movement of cells up a concentration gradient of soluble signals (collectively known as chemoattractants) diffused in solution [14]. Different types of cell migration are recognised based on the type of signals that the cells respond to and the directionality of their movement:Chemokinesis happens when the detection of a chemoattractant by cells causes morphological changes that lead to increased overall motility but no directionality of the movement. Chemokinesis can be induced either in the presence of a concentration gradient or in uniform concentration of a chemoattractant, and therefore constitutes an important control for chemotaxis studies [15,16];Haptotaxis occurs when the chemoattractant gradient is surface bound, for example to the extracellular matrix (ECM) or endothelium [14]. There is an increasing appreciation that leukocyte migration is most likely directed by the mixture of diffused and surface-bound chemoattractants, highlighting the importance of haptotaxis research [14,17];Fugetaxis (chemorepulsion) is defined as the active migration of cells away from the source of the chemokinetic agents, which are, in this case, called chemorepulsants [18];Necrotaxis occurs when necrotic and apoptotic cells release signals that regulate their removal; they can be simultaneously chemoattractive and chemorepulsive to different cell types [19,20].

These definitions of cell migration only describe the relationship between the chemokinetic agent and directionality of cell movement. For the purpose of this review, the term ‘chemotaxis’ will be used synonymously with ‘directional cell migration’, unless otherwise specified.

### 2.2. Types of Specialised Migration: Shape-Dependent

A single migrating cell can be best imagined as a highly polarised entity where a rapid turnover of migratory machinery causes spatial segregation of cellular components within the cell, which, in turn, drives shape change and motion [21]. However, segregation and shape change are not ubiquitous among all migratory cells and we can distinguish three discrete types of motion: amoeboid, mesenchymal and gliding [22]. Gliding motion is unique to keratinocytes and maintains an almost constant shape of cells during motion due to high level of coordination between the leading edge and the rear of the cell [23]. Since it has never been reported in leukocytes, it will not be discussed further. Amoeboid and mesenchymal migration are depicted in Figure 1. Even though they share molecular pathways, they are distinct in how they shape the cells and their movement.

Amoeboid migration is mainly utilised by leukocytes. Its characteristic features include the lack of firm adhesion points and extension of pseudopodia (actin-rich 3D structures). This enables rapid cell migration up to 30 µm/min in in vitro assays [24]. Moreover, the migrating cells preserve the integrity of the extracellular matrix (ECM) that they crawl over [25]. Amoeboid movement is characterised by the ‘hand mirror shape’, where polarisation of the cell causes a defined actin-rich leading edge that is responsible for sensing the environment and mitochondria-rich uropod that anchors the cell and propels it [24] (Figure 1). The leading edge has a high turnover of actin filaments scaffolding that builds the pseudopodia, which participate in the binding of chemokinetic ligands to their receptors [25].

Mesenchymal migration is characteristic of fibroblasts and smooth muscle cells and exclusive to cells of animal lineage [26]. It is much slower than amoeboid migration with cells travelling less than 1 µm/min in in vitro assays [24,26]. During mesenchymal movement, the leading edge and uropod are much less defined, with multiple lamellipodia (2D sheets-like membrane extensions) being extended in all directions (as seen in Figure 1). The hallmarks of mesenchymal movement are the creation of strong adhesion points and the degradation of ECM that accompanies the movement [24,26].

## 3. Macrophage Chemotaxis

Macrophages are phagocytic cells that are involved in the detection of pathogens and tissue damage, removal of apoptotic cells and repair of the tissue after acute inflammatory response. Macrophages rely on migratory responses to perform their functions effectively. They broadly fall into two main categories: tissue-resident (foetal-derived) cells such as Kupffer cells in the liver or microglia in the brain and monocyte-derived cells that infiltrate tissues during inflammation [27]. Regardless of their origin, macrophages are a part of the immune system and, therefore, historically were expected to follow mechanisms similar to other leukocytes. Given that monocytes are direct precursors to tissue-infiltrating macrophages, the two cell types have often been grouped and discussed together, assuming that the migratory machinery is preserved between them [28].

Recent evidence shows that macrophage migration is much more complex than monocytic chemotaxis and does not follow the same pathways. Macrophages have been shown to use a mixture of amoeboid and mesenchymal migration, choosing the optimal mode of action based on the rigidity and porosity of the ECM around them [24,29]. Their speed of migration in vitro was shown to be around 10µm/min, which falls between the reported average speeds for amoeboid and mesenchymal migration [30,31]. The limitations of chemotaxis assays often resulted in this subtle phenotype being underappreciated.

Different chemotaxis assays are appropriate for different research questions and this diversity is reflected in the published macrophage chemotaxis literature. In general, 2D system have been used for molecular and kinetic studies while 3D gel invasion assays have been used to understand transitions between mesenchymal and amoeboid movement and the impact of ECM on macrophage movement [32].

## 4. Methods to Study Macrophage Chemotaxis In Vitro

### 4.1. The ‘Ideal Chemotaxis Assay’

In 1953 and 1954, Harris proposed a set of conditions that should be fulfilled by an ‘ideal chemotaxis assay’ [3,5]. The ‘ideal assay’ that Harris proposed would allow the detection of both chemotaxis and fugetaxis, testing of enhancers and inhibitors of cell migration, as well as precise distinction of different signal-dependent movement types [3]. Harris’ criteria, which were collected by Bignold in 1988, are now considered the gold standard of chemotaxis research [33,34].

According to Harris the ‘ideal chemotaxis assay’ should:Have no passive movement of cells to ensure that any change in the position of the cells is because of its active motion;Ensure that if soluble factors are being tested, they are sufficiently localised in the experimental chamber;Be able to control the concentration gradients of chemoattractants from their source until they reach the cells, especially by preventing any convection currents;Enable cells to move both towards and away from the chemoattractant source in a homogenous environment;Ensure that if a test object is used, it is possible to distinguish between active cell migration and trapping of cells around the given object due to their random movement;Be free from sampling error [3,33,34].

As common-sense and reasonable as those criteria seem, they have proven to be much harder to put into practice than even Harris himself expected. In his in vitro chemotaxis assay for blood leukocytes, published in 1954, he failed to achieve sufficient localisation of soluble factors to his experimental chamber [33]. As pointed out in Table 1, none of the current chemotaxis assays fulfil all of these criteria, and the whole field has adopted the view that ‘the more assays the better’, while largely ignoring the limitations that omitting some of the above points create. While most of the popular assays can detect enhancers and inhibitors of chemotaxis, it has proven much harder to distinguish between chemotaxis, chemokinesis and fugetaxis [34].

Recently, new considerations such as high throughput, real time kinetic data and single-cell vs. population studies became important in choosing and designing chemotaxis assays. In their 2004 review, Frow et al. assessed the most popular assays based on how low-to-high throughput they are versus how much information can be obtained about cell movement behaviour at a single-cell level [34]. Since their goal was to find an optimal method for anti-inflammatory drug discovery projects they favoured the high-throughput–low-information assays as good candidates for drug screens, while noting that transwell assays and under agarose assays provide the middle ground that can be useful for basic science studies [34].

Including these new considerations, the ‘ideal chemotaxis assay’ should, therefore, allow high-throughput studies that can also inform on real-time cell behaviour on a single-cell level (while maintaining the ability to look at whole populations of cells) and also fulfilling Harris’ criteria. The advent of microfluidics, paired with computerised image processing, showed great promise to find this ‘ideal assay’ but it also brought new challenges such as inter-lab reproducibility, rising costs and increasing assay complexity. The current goal for macrophage chemotaxis research is, therefore, finding an assay that can fulfil Harris’ criteria while being relatively straightforward and cheap enough to achieve popularity comparable to transwell assays or under agarose assays.

### 4.2. Transwell Assays—From Boyden Chamber to Real-Time Recordings

The invention of the Boyden chamber in 1962 was a turning point in the chemotaxis field [6]. Despite its limitations, the simplicity of the assay design allowed for well-controlled and reproducible in vitro studies of cell migration and quickly became the staple technique in all cell migration research. As seen in Figure 2, the basic design of a Boyden chamber consists of a well filled with a chemoattractant solution and an insert that gets submerged into it. The inside of the insert (‘top well’) is separated from the well by a porous membrane; therefore, when cells are seeded into the insert, a concentration gradient of a chemoattractant is created by diffusion and cells can migrate through the pores in the membrane to the chemoattractant chamber [6].

One of the main advantages of transwell assays is the ability to clearly distinguish between chemotaxis and chemokinesis, without the need of single-cell tracking, thanks to checkerboard analysis. In checkerboard analysis, a chemoattractant is placed both in the bottom and top wells at varying concentrations to create a range of gradients with different steepness and direction. In checkerboard analysis, increased migration in wells where there is a shallow gradient, or no gradient, is a hallmark of chemokinetic movement, while the lack of migration in shallow gradients accompanied by increased migration in steep chemoattractant gradients signifies chemotactic movement [34].

The Transwell assay design fulfils most of Harris’ criteria (Table 1), except for allowing the cells to migrate away from the source of the chemoattractant. Although there is no precise way to control the gradient, the physical barrier of the porous membrane prevents convection currents from flushing the cells through. In the early experiments, high inter-well variability and relatively high volumes were the main disadvantages of the technique [35]. However, the assay’s simple readout was ideal for studies where the research question was about the overall difference in migratory potential of the whole population rather than differences in their migratory behaviour such as velocity, directionality or mode of movement.

Falk et al. improved the Boyden chamber by creating ChemoTx 96-well plates. They replaced individual inserts with a porous membrane covering the whole plate and placed a drop of the required cell suspension directly over the porous membrane (Figure 2), which required lower volumes [36]. Another major improvement came with the xCELLigence system, which used gold microelectrodes attached to the porous membrane to record real-time migration data by measuring cell impedance of the cells attached to the bottom of the membrane (Figure 2) [37]. To overcome the problem of migrated cells detaching from the membrane, and therefore influencing the reading, IncuCyte created a chemotaxis assay where real-time phase microscopy photos of the top of the membrane are used to measure how many cells have pushed through the membranes instead [28].

Transwell assays have been used to study molecular components of the migratory pathways (thanks to the use of inhibitors and knock-out cells), as well as for drug testing. Single-timepoint transwell systems have been important in discovering modulators of macrophage migration such as leptin and NFκB [38,39]. The advent of real-time recordings added much-needed kinetic data, highlighting that different populations of macrophages have different migratory profiles. Iqbal et al. explored the heterogeneity of migration kinetics between thioglycolate- and biogel-elicited macrophages as well as bone-marrow-derived macrophages and resident peritoneal macrophages [37]. Each population displayed different migratory profiles towards common chemoattractants such as CCL2, CCL3 and CCL5. Given the general ‘one size fits all’ approach in discussing leukocyte migration, the study by Iqbal et al. is an important reminder of the danger of generalising biological processes in heterogenous and diverse cell populations.

Transwell assays can be used to compare two separate populations of cells, however, they cannot look at mixed populations such as whole blood due to the different pore sizes required for different cell types as well as the inability to easily distinguish different cell types once added into the system. Moreover, transwell assays cannot distinguish an inhibitor of chemotaxis from a chemorepellent, since both such agents would result in reduced migration through the membrane. Transwell assays are primarily short-term studies, and therefore unable to investigate processes such as how transendothelial migration of monocytes can influence their maturation into macrophages. Therefore, despite being probably the most used migration assays, transwell systems do not fulfil all of Harris’ criteria of the ‘ideal chemotaxis assay’ and are limited in their capacity to investigate complex migratory behaviours such as cellular decision-making.

### 4.3. Direct Observation and Cell Tracking Chambers—Zigmond and Dunn Chambers

The Zigmond chamber was developed in 1977 with the aim of improving the stability of soluble chemoattractant gradients in comparison to transwell assays [40]. Sometimes called an orientation chamber, it consists of a slide with two channels separated by a raised barrier. When one channel contains pure buffer and the other contains chemoattractant solution, the narrow passage that the barrier creates supports the formation of a stable chemoattractant gradient. Cells of interest are attached to a cover slip that is inverted over the wells and the passage so that cells can sense the gradient and migrate in any direction on the cover slip; their movement can be recorded.

Despite theoretically creating a stable gradient, the fact that the Zigmond chamber was an open system in practice meant that convection currents were created in the system. To overcome that problem, Zicha and Dunn created a novel closed chemotaxis system in 1991 [41]. They replaced the channels with two concentric wells separated by a raised barrier. The circular design meant that inverting a coverslip with the cells over it sealed the system, making it a closed circuit, and therefore ensuring the stability of the gradient. They demonstrated gradient stability for up to 30 h by both mathematical modelling and experiments with dyes [41].

Wheeler et al. used a Dunn chamber to investigate the morphology of migrating bone-marrow-derived macrophages [42]. Using this system, alongside transwell assays, they were able to show that members of Rho GTPases family Rac1 and Rac2 are responsible for the formation of lamellipodia and podosomes, but not migration itself. Using Dunn chambers, they were, therefore, able to demonstrate redundancy in macrophage migratory machinery, which can, in part, explain their diverse movement [42].

The Dunn chamber is impractical to use for testing chemotaxis inhibitors and enhancers. Moreover, the system is a low-throughput one for pharmacological studies and is often used alongside more efficient systems such as transwell assays.

### 4.4. ‘Under Agarose’ Migration Assay

The basic principle behind under agarose assay is that a cell-impermeable agarose gel is laid over a glass or plastic slide and a row of three wells are punched out in the gel. Cells are seeded in the middle well, while small volumes of buffer or chemoattractant solution are added to the side wells. Solutions diffuse through the agarose gel to create a chemoattractant gradient [43]. Cells can then detect the chemoattractant and move up the concentration gradient under the agarose gel. The assay is typically single-timepoint—at the end of the experiment, the agarose gel is removed, cells are fixed and stained on the slide and the distribution pattern is used to quantify the movement [43]. Popular quantification methods include counting the number of cells that left the middle well or dividing the distance between the wells into serial ‘slices’ and looking at the distribution of cells across the slices.

Despite theoretically having the potential to include multiple wells, in practice, concentration gradients are difficult to reliably control in multiple comparison setting. Under agarose, migration fulfils most of Harris’ criteria, however, it is rarely used for testing modulators of migration as it is a low-throughput method [34]. Moreover, chemokinetic agents may not be detected in this system because the increase in random movement without the incentive to migrate under the agarose may result in increased movement only within the well, which would not be detected in a single-timepoint assay.

### 4.5. TAXIScan and Ibidi µ-Slides—Movement towards Microfluidics

As previously mentioned, microfluidics’ chemotaxis assays bring promise to fulfil all the criteria of the ‘ideal chemotaxis assay’, however, they tend to be hard to reproduce and are not yet commercially available. TAXIScan and ibidi tried to overcome this limitation by introducing commercially available, real-time chemotaxis assays based on simplified microfluidics designs. Even though TAXIScan is no longer commercially available, it was included in this review because of the important contributions to the field that this technology enabled.

TAXIScan consists of a small metal chip with 12 wells, connected in pairs by narrow (8 µm wide) channels [44]. Cells are injected into one of the wells while the chemoattractant is injected into the second well. When cells migrate into the channel and towards the chemoattractant, their movement is recorded in real-time using phase microscopy. Ibidi µ-slides operate on a similar principle, where two triangular reservoirs are connected by a narrow observation area that contains cells of interest [45]. However, the advantage of ibidi µ-slides lies in the ability to seed the observation area with an ECM matrix, creating a 3D assay, as well as the ability to seed cells as a source chemoattractant into one of the reservoirs.

Both systems have reported stable concentration gradients for up to 48 h, however, they also both rely on passive diffusion of the gradient. The main difference lies in the fact that TAXIScan’s narrow channel allows, effectively, only one cell to pass at one time, while ibidi’s observation area is 1 mm wide, and therefore allows unconstrained cell passage. Since macrophage movement was shown to depend on the constrictions of their extracellular environment, the lack of a narrow channel may be advantageous [32]. The main disadvantage of both systems is the cost and requirements for specialised equipment to perform the experiments.

Vogel et al. used TAXIScan to investigate differences in migration and adherence between non-polarised and M1/M2 polarised monocyte-derived primary human macrophages [44]. They were able to show that non-polarised and M2 anti-inflammatory macrophages could migrate further than M1 macrophages. They attributed the difference in migratory capacity to non-polarised and M2 macrophages being able to rearrange their cytoskeleton more efficiently and forming multiple filopodia, while M1 macrophages remained more spherical in shape, resulting in worse migration [44].

### 4.6. Microfluidics Chemotaxis Assays

Microfluidics chemotaxis devices are usually generated when a 3D-printed plastic construct, typically made of polydimethylsiloxane (PDMS), is attached to a glass or plastic slide, and the movement of cells through the system is recorded in real time [46]. Their main advantage is the ability to design and print the systems specifically to test a given hypothesis. In practice, microfluidic devices are not easily manipulated, and each study uses a unique design. Microfluidic circuits rely mainly on the passive diffusion of chemoattractant solution through the system to establish the gradient. However, complex maze-like structures, where the chemoattractant solution is mixed with pure buffer, are used to control the steepness of the gradient at different points of the circuits and expose cells to areas of different intensity gradients [47].

The active flow of a chemoattractant solutions is also possible, which provides the most stable exposure to chemoattractants for the cells. Both passive mazes and active flow give better control over the gradient, but may be prone to convection currents. Microfluidic circuits can be used to solve more complex questions about cell migration, for the first time giving the researchers a platform that can be modulated to answer specific scientific questions rather than having to find a research question that fits the assay. Because of that, they can be designed to fulfil Harris’ criteria as well as incorporate Frow’s new requirements. In practice, however, microfluidics is still in its infancy. Most laboratories do not have the resources and expertise to design and print their own circuits, while the published ones are not commercially available. Moreover, PDMS has been shown to be cytotoxic, which raises questions about the impact of the system itself on the cell biology [46].

Most of the current chemotaxis research in microfluidic systems has been done on dendritic cells (DCs), which have very defined and well-understood responses to the CCL19/CCL21 chemotaxis axis [17]. Research questions studied in DCs are relevant to macrophage research, such as cellular decision-making during migration in heterogenous environments with different pore sizes [48]. Even though such questions have been studied in 3D gel migration assays, the carefully designed microfluidics systems which combine different widths of the channel in one system have a clear advantage over comparing the same cell population in different gels with different (worse controlled) porosity [32,48].

### 4.7. Gel Invasion—In Vitro 3D Migration Assay

Macrophage migration in vivo happens almost exclusively in 3D tissue environments; therefore, it can be argued that all 2D chemotaxis systems have limited physiological relevance. Collagen, fibrin or ECM mixtures (such as Matrigel) gels are used to create 3D migration environments with different rigidity and porosity, with the aim of mimicking different types of tissue [32]. Chemoattractants can diffuse through the gels either directly from a reservoir or by setting up adjacent gels. They can also be seeded in the top chamber of a transwell assay [29]. Cells can be either fixed and counted at a single timepoint or monitored continuously. As the gels are translucent, cell movement through the matrix can be recorded and analysed for velocity, displacement, directionality and cell shape changes. Another common method of quantification is seeding the cells on top of the gel and counting how many of them enter the matrix.

Despite having the advantage of a 3D environment, gel invasion assays have a limited ability to answer basic biological questions because of the impact of the artificial matrix itself on cell movement. Cui et al. used 3D gel invasion as a single-timepoint assay to investigate differences between peritoneal macrophage polarisation states. They concluded that the difference lies in the expression of CD11a, CD11b, CD11c and CD11d integrins and their adhesion to the ECM [29]. Their study showed the ability to use 3D in vitro systems for molecular studies that inform not only on the molecular pathways involved in migration but also the shape changes that macrophages undergo and their ability to switch between amoeboid and mesenchymal migration. Pakshir et al. demonstrated the that 3D gel invasion assays can be also used to study cell-to-cell interactions in real time [49]. They successfully implemented videomicroscopy to show that macrophage directional migration in a 3D matrix can be induced by mechanosensing of physical changes that contractile fibroblasts create in the ECM, independently of soluble chemotattractants.

## 5. Conclusions

Macrophage chemotaxis is unique among all leukocytes in requiring a complex interplay between amoeboid and mesenchymal migratory mechanisms. The diverse functions of macrophages in tissues may have created a need for a more plastic migratory mechanism adaptable to different tissue conditions. Due to the limitations of current in vitro chemotaxis assays, leukocyte migration has been studied more extensively on polymorphonuclear cells and left the unique nature of macrophage migration underappreciated for a long time. With the advent of real-time systems, microfluidics and better imaging techniques, we are gaining a better understanding of the interplay between the amoeboid and mesenchymal migration. However, all chemotaxis assays rely purely on soluble chemoattractant gradients, while cells in vivo likely encounter a mixture of soluble and surface-bound chemoattractants. While attempts have been made to create easy and robust haptotaxis assays, further developments in this area will be crucial in understanding macrophage migration [14,17].

Many questions remain to be addressed. What causes the heterogeneity in macrophage mode of movement, and is it purely driven by the external environment of the cells? How do leukocytes decide on the most efficient mode of migration in a multi-cue environment? Are soluble and surface-bound gradients equally good and equally important in directing macrophage movement? Additionally, perhaps most importantly, can we modulate the unique migratory mechanisms of monocytes and macrophages for a therapeutic advantage in diseases such as atherosclerosis, where myeloid cell migration plays crucial roles in pathogenesis?

## Figures and Tables

**Figure 1 biology-09-00439-f001:**
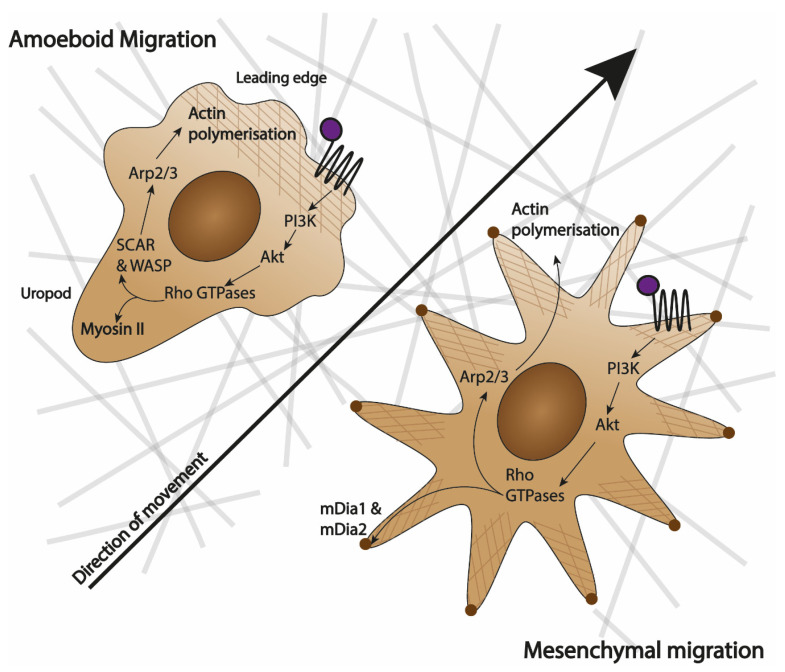
Similarities and differences between amoeboid and mesenchymal movement. Amoeboid migration is characterised by a ‘hand-mirror’ cell shape with a clear leading edge and a lagging uropod, whilst mesenchymal migration does not have a defined shape and is characterised by multiple lamellipodia. Amoeboid cells do not degrade extracellular matrix (ECM) that they move on (represented by the grey lines), while mesenchymal cells do. When a chemoattractant attaches to a receptor (purple dot attaching to 7 transmembrane domains), in both cases, it activates a PI3K/Akt cascade that leads to Rho GTPases. In amoeboid cells, Rho GTPases either directly activate Myosin II in the uropod or cause actin (represented by brown grid) polymerisation at the leading edge by activating SCAR and WASP proteins, and subsequently Arp2/3 [21,24]. In mesenchymal cells, Rho GTPases either activate formins mDia1 and mDia2 to create adhesion points (brown dots) to the ECM or acting via Arp2/3 cause actin polymerisation that lead to the extending of lamellipodia [21].

**Figure 2 biology-09-00439-f002:**
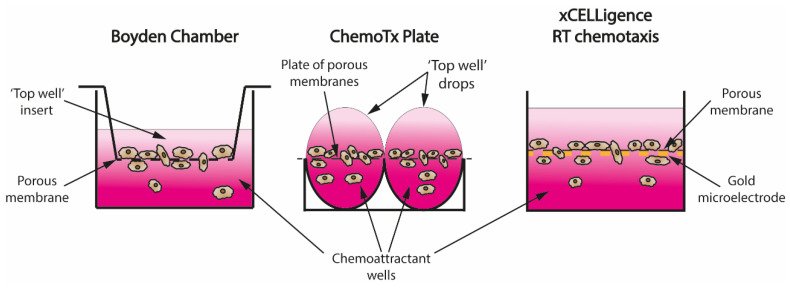
Evolution of the transwell assay design. (**left**) The original Boyden Chamber, created in 1962, consisted of a well-filled chemoattractant solution into which a ‘top well’ insert was placed with cells seeded in pure cell media. Due to simple diffusion, a chemoattractant gradient was created (represented by pink gradient) that induced directional migration of cells through the pores into the chemoattractant well; (**middle**) ChemoTx plates aimed at reducing high inter-well variability and high volumes of solution required with Boyden chamber by replacing individual inserts with one big porous membrane overlaying the entire 96 well plate; drops of cell suspension were placed directly onto the membrane over the chemoattractant wells to create a gradient via simple diffusion and allow cells to migrate through the pores; (**right**) xCELLigence Real Time Chemotaxis assay allowed collection of real-time kinetic data for the first time in transwell assays by attaching a gold microelectrode to the bottom of the porous membrane. Migration was quantified by measuring cell impedance of the cells that migrated through the pores and attached to the bottom of the membrane.

**Table 1 biology-09-00439-t001:** Comparison of popular chemotaxis assays based on Harris’ criteria and desirable features for current needs in chemotaxis research. Inspired by Frow et al. (2004) [34].

	Transwell Assays	Direct Observation and Cell Tracking Chambers	Under Agarose Migration Assay	TAXIScan and Iibidi µ-Slides	Microfluidics	Gel Invasion
Controlled stable gradient?	**+**	**+/−**	**−**	**+**	**+**	**−**
Fugetaxis detection or reversibility?	**−**	**+**	**+**	**−**	**+/−**	**+/−**
Distinction between chemotaxis and chemokinesis?	**+**	**+**	**−**	**+/−**	**+**	**+**
Single cell tracking?	**−**	**+**	**−**	**+**	**+**	**+**
Parallel screening of multiple conditions?	**−**	**−**	**+/−**	**+**	**+**	**−**
High throughput?	**+/−**	**−**	**−**	**−**	**+/−**	**−**
Real-time recording?	**+/−**	**+**	**−**	**+**	**+**	**+/−**
Specialised equipment needed?	**+/−**	**−**	**−**	**+**	**+**	**−**

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
