# Peer review of "How Have Leukocyte In Vitro Chemotaxis Assays Shaped Our Ideas about Macrophage Migration?"

_biology, 2020, doi:10.3390/biology9120439_

Round 1
Reviewer 1 Report
The paper written by Rumianek and Greaves is a nice overview about the methology of chemotactic assays. It not only summarizes the methodology, but gives a nice introduction about chemotactic migration, the terms used in the field, and then summarizes the methodologies themselves giving examples, how a certain methodology pushed further the field in our understanding how macrophages move. I suggest to accept the review without any change.
Author Response
We thank Reviewer 1 for their kind comments on our review.
Reviewer 2 Report
This is a very complete and well-written review about different assays to evaluate macrophage migration.
Author Response
We thank Reviewer 2 for their kind comments on our review.
Reviewer 3 Report
This well-written review article describes the strengths and weaknesses of the main methods used to quantify and study chemotaxis. The manuscript is generally thorough and clear and I appreciated the historical perspective of how the methodologies have evolved. I think the manuscript is a useful contribution to the field and have only two relatively minor comments.
- The title suggests that the review focuses on macrophages specifically, but the assays described are used with neutrophils as well as other cell types. thus, although some limitations may be macrophage-specific, overall the strengths and weaknesses apply to other cell types as well. For this reason, a more general title may ensure that the article reaches a majority of interested readers.
- This is a minor point, but the company that manufactured the EZ TAXIScan drowed (literally) in the Fukushima tusnami, thus this instrument is not longer available, which is unfortunate. We are using the one in my lab until it breaks, but it may be useful to readers to mention that this instrument is no longer being manufactured.
Author Response
We thank Reviewer 3 for their thoughtful comments and suggestions.
In point 1 the reviewer makes a good suggestion. We have changed the title of the review to include the word leukocyte to ensure that the review will reach a broader audience.
In point 2 we have added a footnote to explain that TAXIScan is no longer commercially available (bottom of page 8).
Reviewer 4 Report
In the following review the authors described exclusively the classical and modern chemotaxis tools susceptible for the analysis of macrophage migration pointing out their advantages and disadvantages. Overall, the aspect and the concept and of the manuscript are systemic and well written. However, the last paragraph (4.4.) concerning the 2D and 3D chemotaxis might be enriched by inserting additional examples of macrophages migration that cover time-lapse imaging (etc.).
Author Response
Following Reviewer 4's suggestion we have now added text to the last paragraph in the section 'Gel invasion – in vitro 3D migration assay' to include a recent example of videomicroscopy used in 3D gel invasion assays.
We thank the Reviewer 4 for their thoughtful comments on our review.